# Toward Unpaired Multi-modal Medical Image Segmentation via Learning Structured Semantic Consistency

**Jie Yang**[1]                                                                                        YANGJIE5@LINK.CUHK.EDU.CN
**Ye Zhu**[1]                                                                                               ZHUYE1@CUHK.EDU.CN
**Chaoqun Wang**[1]                                                                     CHAOQUNWANG@LINK.CUHK.EDU.CN
**Zhen Li**[1]                                                                                     LIZHEN@LINK.CUHK.EDU.CN
**Ruimao Zhang**[1,*]                                                                         RUIMAO.ZHANG@IEEE.ORG
[1] *Shenzhen Research Institute of Big Data, The Chinese University of Hong Kong, Shenzhen*
[*] *Corresponding author*

**Editors:** Accepted for publication at MIDL 2023

## Abstract

Integrating multi-modal data to promote medical image analysis has recently gained great attention. This paper presents a novel scheme to learn the mutual benefits of different modalities to achieve better segmentation results for unpaired multi-modal medical images. Our approach tackles two critical issues of this task from a practical perspective: (1) how to effectively learn the semantic consistencies of various modalities (e.g., CT and MRI), and (2) how to leverage the above consistencies to regularize the network learning while preserving its simplicity. To address (1), we leverage a carefully designed External Attention Module (EAM) to align semantic class representations and their correlations of different modalities. To solve (2), the proposed EAM is designed as an external plug-and-play one, which can be discarded once the model is optimized. We have demonstrated the effectiveness of the proposed method on two medical image segmentation scenarios: (1) cardiac structure segmentation, and (2) abdominal multi-organ segmentation. Extensive results show that the proposed method outperforms its counterparts by a wide margin.

**Keywords:** Unpaired multi-modal learning, Structured semantic consistency learning, Medical image segmentation

## 1. Introduction

Assessing specific diseases often involves using different imaging modalities, such as CT and MRI, which provide distinct information on tissue structure. In clinical practice, these modalities are combined to achieve a comprehensive understanding of organs for disease assessment and treatment planning (Karim et al., 2018; Cao et al., 2017). Despite the differences in appearance between CT and MRI data, similar techniques like quantitative segmentation are crucial for diagnosis (Dou et al., 2020). Previous research primarily focused on developing robust segmentation models for single-modality applications (Cao et al., 2021; Gao et al., 2021). However, due to domain shifts between modalities, models trained on one modality often fail when applied to another, posing challenges for real-world clinical analysis (Chen et al., 2020).

In the literature, some recent studies (Wang et al., 2021b; Cheng et al., 2022) have been presented to address the aforementioned issue via joint representation learning from

multi-modalities. However, this joint representation learning principally necessitates spatial alignment and co-registered sequences within multi-modalities, *e.g.,* multi-sequence MRI (T1, T1c, T2, FLAIR). For the unpaired multi-modal data, *e.g.,* CT and MRI, such a scheme is infeasible because of the spatial misalignment. Recently, Valindria *et al.* (Valindria et al., 2018) proposed four kinds of dual-steam CNNs to alleviate the negative domain shift between unpaired CT and MRI, where assigning modalities with their specific feature extractors greatly affects the model's parameter efficiency and limits the model's ability to handle more modalities. Dou *et al.* (Dou et al., 2020) further designed both modality-specific and modality-shared modules to accommodate the appearance variance of different modalities. Despite significant efforts to pursue multi-modal medical image segmentation, it still poses some challenges for real-world applications due to the following issues. **First**, discovering how to fully explore the semantic associations of multiple modalities is critical but very difficult because there is no pixel-to-pixel correspondence in the unpaired input images in practice. **Second**, how to discard complex model design and leverage the above semantic associations to regularize the network learning while preserving its simplicity remains intractable nowadays.

To address the above issues, this paper presents a novel method for performing unpaired multi-modal medical image segmentation based on a single Transformer by learning the structured semantic consistency between modalities, *i.e.* the consistencies of semantic class representations and their correlations. Specifically, the unpaired multi-modal medical images, *e.g.,* CT and MRI, are firstly fed into a shared Transformer backbone to extract multi-scale feature representations. For each modality, we further introduce a set of modality-specific class embeddings, each of which indicates a global representation of one semantic class. It is updated during the training phase to learn the specific class representation across the entire dataset. In practice, these modality-specific class embeddings are learnable and fed into an elaborate External Attention Module (EAM) to interact with the feature maps of the corresponding modal images. By doing this, the image-specific class embeddings and their correlations of a certain image can be further extracted. Furthermore, structured semantic consistency across modalities can be achieved gradually by implementing consistency regularizations at the modality-level and image-level respectively. During the testing phase, we discard all EAMs and only hold a single Transformer for predicting the segmentation results of various modalities.

In summary, the main contributions of this paper are as follows: **(1)** We propose a novel method to learn to segment multi-modal medical images by using a single Transformer backbone. **(2)** We introduce a plug-and-play External Attention Module to assist the backbone in discovering the semantic association and learning the structured semantic consistency by using unpaired multi-modal medical images. **(3)** We evaluate our method on two different multi-class segmentation tasks with 2D and 3D configurations, showing the effectiveness of our method in various settings.

## 2. Methodology

### 2.1. Problem Setting and Framework Overview.

Considering two unpaired medical images $\{\mathbf{X}_{M_1}, \mathbf{X}_{M_2}\}$ extracted from different modalities and their corresponding label maps $\{\mathbf{Y}_{M_1}, \mathbf{Y}_{M_2}\}$, the overall framework of the proposed

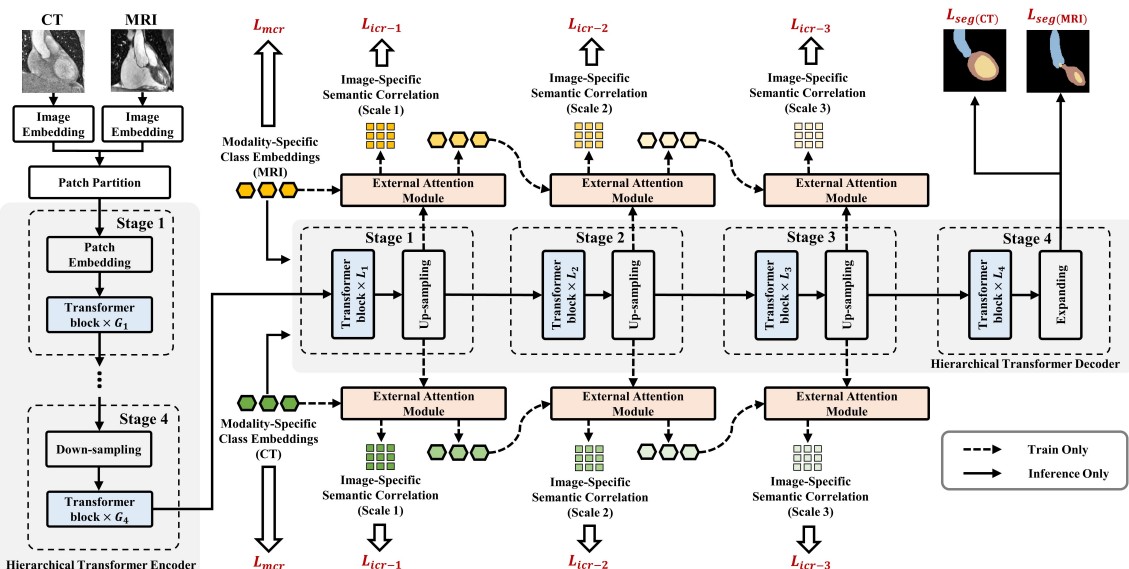

Figure 1: Overview of our proposed unpaired multi-modal medical image segmentation framework via single Transformer architecture and the proposed EAMs. All parts linked by dotted lines can be removed during the inference phase.

method is depicted in Fig. 1. For simplicity, we use the 2D model as an illustration which can be easily extended to the 3D model. We first feed three consecutive slices as the inputs $\{\mathbf{X}_{M_1}, \mathbf{X}_{M_2}\} \in \mathbb{R}^{3 \times H \times W}$ into the modality-specific image embedding module, which is simply implemented by two consecutive $1 \times 1$ convolutional layers, making the resolution and dimension of the inputs unchanged. The embedded feature maps from the two modalities are then fed into a single Unet-shaped Transformer-based segmentation network that includes encoder and decoder subnetworks for pixel-wise dense prediction. Specifically, the decoders could generate multi-scale features that are combined with the encoders' features, *i.e.* $2C \times \frac{H}{P} \times \frac{W}{P}$ at Stage 1, $2C \times \frac{H}{2P} \times \frac{W}{2P}$ at Stage 2, $4C \times \frac{H}{4P} \times \frac{W}{4P}$ Stage 3, and $8C \times \frac{H}{8P} \times \frac{W}{8P}$ at Stage 4, where $P$ is the patch size in Transformer and is 4 by default.

To align the modalities during the training phase, we explore two kinds of semantic information, termed ***modality-specific class embeddings*** and ***image-specific semantic correlations***. The former is a set of learnable vectors for each modality, each of which presents one semantic class, *e.g.,* `Liver` or `Spleen`. It aims to learn the global class representations of each modality. The latter is used to present the inter-class relationships within a specific image. In practice, a newly designed ***External Attention Module*** (**EAM**) is introduced to update the above learnable class embeddings from modality-specific to image-specific, and extract the semantic correlations of a specific image at multiple scales, as shown in Fig. 1.

We explicitly facilitate the consistency of the two modalities' representations. 1) We firstly introduce the global consistency regularization $\mathcal{L}_{mcr}$ to minimize the representation distance between modality-specific class embeddings. It aims to globally align the semantic class representations of two modalities. Such consistency will also implicitly affect the pixel-level representation learning of each sample since these modality-specific class embeddings also interact with corresponding images of each modality through the training process. 2)

We further align semantic correlations between two modalities at image-level by minimizing $\mathcal{L}_{icr}$ (*i.e.* symmetrical Kullback-Leibler divergence). Such a scheme allows generating many sample pairs to drive semantic correlation alignment, which makes the optimized model more robust to sample variation.

## 2.2. External Attention Module

For simplicity, we illustrate the proposed EAM using feature maps $\mathbf{F} \in \mathbb{R}^{\frac{H}{4P} \times \frac{W}{4P} \times 4C}$ at Scale-1 and such mechanism can be applied to other scales as well, as shown in **Appendix** 2.

**Class Embeddings.** In practice, the learnable modality-specific class embeddings are used to learn the class representations of a specific modality across the entire dataset, while each image should have its own class representations that differ from the global ones due to their appearance variance. Thus, we employ the Cross-Attention Mechanism (CA) for updating class embeddings by interacting with multi-scale feature maps of a certain image to generate its image-specific class embeddings. Specifically, we utilize $\mathbf{F}$ to calculate the key and value of CA by linear projection, while the query of CA is calculated employing modality-specific class embeddings $\mathbf{Q} \in \mathbb{R}^{Z \times 4C}$ as follows:

$$\mathbf{q} = \mathbf{Q}\mathbf{W}_Q, \ \mathbf{k} = \mathbf{F}\mathbf{W}_K, \ \mathbf{v} = \mathbf{F}\mathbf{W}_V, \tag{1}$$

$$\mathrm{CA}(\mathbf{Q}, \mathbf{F}) = \mathrm{softmax}(\frac{\mathbf{q}\mathbf{k}^{\mathsf{T}}}{\sqrt{d}})\mathbf{v}, \tag{2}$$

where $\mathbf{W}_K$, $\mathbf{W}_V$, $\mathbf{W}_Q \in \mathbb{R}^{4C \times 4C'}$ are the parameter matrices for linear projection. $d$ is the channel dimension of $\mathbf{q} \in \mathbb{R}^{Z \times 4C'}$ and $\mathbf{k} \in \mathbb{R}^{(\frac{H}{4P} \times \frac{W}{4P}) \times 4C'}$. The softmax($\cdot$) denotes the softmax function along the spatial dimension. The $\mathbf{q}\mathbf{k}^{\mathsf{T}} \in \mathbb{R}^{Z \times \frac{H}{4P} \times \frac{W}{4P}}$ indicates the *Semantic-aware Feature Maps* extracted from a single CA head at Scale-1, where $Z$ denotes the total numbers of classes. The Multi-head Cross Attention (MCA) is the extension with $N$ independent CAs and project their concatenated outputs as follows:

$$\mathrm{MCA}(\mathbf{Q}, \mathbf{F}) = \mathbb{C}( \ \mathrm{CA}_1(\mathbf{Q}, \mathbf{F}), ..., \mathrm{CA}_N(\mathbf{Q}, \mathbf{F}) \ ) \ \mathbf{W}_O, \tag{3}$$

where $\mathbb{C}$ denotes the concatenation operation. $\mathbf{W}_O \in \mathbb{R}^{4C' \times 4C}$ is the learnable parameter matrix, and we have $4C' = 4C/N$. Here the *Semantic-aware Feature Maps* extracted from multiple attention heads at Scale-1 can be presented as $\mathbf{A}_1 \in \mathbb{R}^{Z \times N \times \frac{H}{4P} \times \frac{W}{4P}}$, which can be further adopted to derive image-specific semantic correlations. In this way, the $\mathbf{Q}$ can be updated by:

$$\hat{\mathbf{Q}} = \mathrm{MCA}( \ \mathrm{Norm}(\mathbf{Q}), \ \mathrm{Norm}(\mathbf{F}) \ ) + \mathbf{Q}, \tag{4}$$

$$\tilde{\mathbf{Q}} = \mathrm{MLP}( \ \mathrm{Norm}(\hat{\mathbf{Q}}) \ ) + \hat{\mathbf{Q}}, \tag{5}$$

where $\tilde{\mathbf{Q}} \in \mathbb{R}^{Z \times 4C}$ reflects image-specific class representations by collecting image-specific semantic information from feature maps of a particular modal image. In practice, the $1 \times 1$ convolution operation is further used to reduce the dimension of the above $\tilde{\mathbf{Q}}$ to $Z \times 2C$ and obtain image-specific class embeddings $\mathbf{Q}_1$ for the next scale of updates. The operations at the other scales are identical to those described above, and the only difference is that we adopt image-specific class embeddings (*e.g.* $\mathbf{Q}_1$ or $\mathbf{Q}_2$) to replace $\mathbf{Q}$ in the above equations.

**Semantic Correlations.** The semantic correlations reflect the inter-class representation similarities. In practice, we propose a three-step process to extract the semantic correlation matrix in a specific image. Again, we explain the corresponding operations at Scale-1 as an illustration. Given the modality-specific class embeddings $\mathbf{Q}$ and multi-head semantic-aware feature maps $\mathbf{A}_1$, the semantic correlations $\mathbf{E}_1$ can be calculated via semantic filtering, semantic re-weighting, and semantic aggregation, respectively. The operations at the other scales are the same, the only difference is that we replace $\mathbf{Q}$ with image-specific class embeddings $\mathbf{Q}_1$ and $\mathbf{Q}_2$ to calculate $\mathbf{E}_2$ and $\mathbf{E}_3$.

**1) Semantic Filtering.** Given a particular semantic class embeddings in $\mathbf{Q}$, the purpose of such a step is to calculate its relevance to all semantic classes at the token level. Since $\mathbf{A}_1 \in \mathbb{R}^{Z \times N \times (\frac{H}{4P} \times \frac{W}{4P})}$, the dimension of tokens is $Z \times N$. We can divide the features of tokens into $Z$ groups, each of which is corresponding to a particular semantic class. Thus we have $\mathbf{A}_1 = \{\mathbf{A}_1^j\}_{j=1}^Z$ and $\mathbf{A}_1^j \in \mathbb{R}^{N \times \frac{H}{4P} \times \frac{W}{4P}}$. Similarly, we can rewrite $\mathbf{Q}$ as $\{\mathbf{Q}^i\}_{i=1}^Z$ and $\mathbf{Q}^i \in \mathbb{R}^{4C}$. For $i$-th class, we generate the semantic kernel as $\mathbf{K}^i = \mathbf{Q}^i \mathbf{W}^i$, where $\mathbf{W}^i \in \mathbb{R}^{4C \times N}$ is the parameter matrix that is corresponding to the $i$-th class. To calculate the similarity between $i$-th and $j$-th classes at token level, we can directly reshape $\mathbf{K}^i$ into $\mathbb{R}^{N \times 1 \times 1}$ to perform filtering on $\mathbf{A}_1^j$ as follows,

$$\mathbf{S}_1^{ij} = \mathbb{F}(\mathbf{K}^i, \mathbf{A}_1^j), \tag{6}$$

where the function $\mathbb{F}(\cdot, \cdot)$ denotes the convolutional operation, and $\mathbf{S}_1^{ij} \in \mathbb{R}^{\frac{H}{4P} \times \frac{W}{4P}}$ is a similarity map of $i$-th class for the $j$-th class, where tokens with higher response scores indicate the higher correlation to $i$-th class representation in $\mathbf{Q}$. That is reasonable in practice. For example, the `left kidneys` and `right kidneys` should have similar structural representations due to their similar appearance, shape, and size. Thus, the class filter of `left kidneys` should also have the highest response to the region of the `right kidneys`.

**2) Semantic Re-weighting.** Given similarity maps between $i$-th class representation in $\mathbf{Q}$ and all groups' feature maps of $\mathbf{A}_1$, then we have $\mathbf{S}_1^i = \{\mathbf{S}_1^{i1}, ..., \mathbf{S}_1^{ij}, ..., \mathbf{S}_1^{iZ}\} \in \mathbb{R}^{Z \times (\frac{H}{4P} \times \frac{W}{4P})}$. By conducting the softmax operation on each spatial position of $\mathbf{S}_1^i$, each element in $\mathbf{A}_1$ is weighted by the gating function as follows,

$$\mathbf{B}_1^i = \mathbf{A}_1 \odot \mathbb{B}(\sigma(\mathbf{S}_1^i)), \tag{7}$$

where $\sigma(\cdot)$ is softmax operation, $\mathbb{B}$ is the broadcast operation to extend the dimension of input to $Z \times N \times \frac{H}{4P} \times \frac{W}{4P}$, and $\odot$ denotes the element-wise multiplication. In this way, we obtain $\mathbf{B}_1^i = \{\mathbf{B}_1^{i1}, ..., \mathbf{B}_1^{ij}, ..., \mathbf{B}_1^{iZ}\} \in \mathbb{R}^{Z \times N \times (\frac{H}{4P} \times \frac{W}{4P})}$, where $\mathbf{B}_1^{ij}$ denotes the correlation map between $i$-th class representation in $\mathbf{Q}$ and $j$-th class feature maps.

**3) Semantic Aggregation.** To generate the final correlation map, we conduct the normalized summation on $\mathbf{B}_1^i$ along the last three dimensions to realize semantic aggregation,

$$\mathbf{E}_1^i = \frac{\sum_{(N, \frac{H}{4P}, \frac{W}{4P})} \mathbf{B}_1^i}{\sum_{(\frac{H}{4P}, \frac{W}{4P})} \sigma(\mathbf{S}_1^i)} \tag{8}$$

where $\mathbf{E}_1^i \in \mathbb{R}^Z$ is the normalized correlation vector, and each element in $\mathbf{E}_1^i$ presents the relevance between the $i$-th class representation in $\mathbf{Q}$ and one specific class representation of

the input image. Finally, image-specific semantic correlations at Scale-1 can be presented as $\mathbf{E}_1 = \{\mathbf{E}_1^1, ..., \mathbf{E}_1^i, ..., \mathbf{E}_1^Z\} \in \mathbb{R}^{Z \times Z}$. The operations of EAM at other scales are the same as Scale-1's, but with different input feature dimensions. In our scheme, the proposed EAM outputs the above semantic correlations at Scale-1 for each modality, *i.e.* denoted by $\mathbf{E}_{1:M_1}$ and $\mathbf{E}_{1:M_2}$ for CT and MRI, which can be continuously updated during the training process. We motivate to dynamically align the semantic correlations of images from different modalities in the training phase since we intuitively assume that the inter-class relationships shown in CT are still valid in MRI.

## 2.3. Objective Functions

**Auxiliary Prediction Loss.** We introduce an auxiliary loss to supervise the semantic prediction of each pixel on multi-scale semantic-aware feature maps by using cross-entropy (CE) and Dice loss (DSC),

$$\mathcal{L}_{\text{aux}} = \sum_{\lambda} \Big[ \text{CE}(\mathbb{F}_{1 \times 1}(\mathbf{A}_\lambda), \mathbf{Y}_\lambda) + \text{DSC}(\mathbb{F}_{1 \times 1}(\mathbf{A}_\lambda), \mathbf{Y}_\lambda) \Big], \tag{9}$$

where $\lambda \in \{1, 2, 3\}$ represents three scales, and $\mathbb{F}_{1 \times 1}(\cdot)$ denotes the function with a $1 \times 1$ convolution operation to reduce the dimension of multi-head.

**Modality-level Consistency Regularization.** We introduce consistency regularization to globally align the class representations of two modalities. Let $\mathbf{Q}_{M_1}$ and $\mathbf{Q}_{M_2} \in \mathbb{R}^{Z \times 4C}$ denote modality-specific class embeddings of two modalities. Then the modality-level consistency regularization can be presented as follows:

$$\mathcal{L}_{\text{mcr}} = \sum_{i=1}^{Z} (1 - \frac{\mathbf{Q}_{M_1}^{i\mathsf{T}} \mathbf{Q}_{M_2}^i}{\|\mathbf{Q}_{M_1}^i\| \cdot \|\mathbf{Q}_{M_2}^i\|}), \tag{10}$$

where $Z$ denotes the total number of semantic classes.

**Image-level Consistency Regularization.** We further utilize the symmetrical Kullback-Leibler (KL) divergence to locally align image-specific semantic correlations of each modality. For two modalities $M_1$ and $M_2$, let $\mathbf{E}_{\lambda:M_1}^i$ and $\mathbf{E}_{\lambda:M_2}^i$ denote correlation vectors corresponding to class $i$-th at Scale-$\lambda$. The image-level consistency regularization can be presented as follows:

$$\mathcal{L}_{\text{icr}} = \sum_{\lambda} \sum_{i=1}^{Z} \Big[ \mathcal{D}_{\text{KL}}(\sigma(\mathbf{E}_{\lambda:M_1}^i / \tau) \parallel \sigma(\mathbf{E}_{\lambda:M_2}^i / \tau)) \\ + \mathcal{D}_{\text{KL}}(\sigma(\mathbf{E}_{\lambda:M_2}^i / \tau) \parallel \sigma(\mathbf{E}_{\lambda:M_1}^i / \tau)) \Big], \tag{11}$$

where $\mathcal{D}_{\text{KL}}(\cdot \parallel \cdot)$ denotes the relative entropy. The $\sigma(\cdot)$ denotes the softmax operation along the class dimension. $\tau$ is a temperature hyper-parameter to control the softness.

**Overall Objective Function.** The overall objective function of the proposed method can be presented as follows:

$$\mathcal{L} = \mathcal{L}_{\text{seg}}^{M_1} + \mathcal{L}_{\text{seg}}^{M_2} + \alpha(\mathcal{L}_{\text{aux}}^{M_1} + \mathcal{L}_{\text{aux}}^{M_2}) + \beta \mathcal{L}_{\text{mcr}} + \gamma \mathcal{L}_{\text{icr}}, \tag{12}$$

where $\mathcal{L}_{\text{seg}}^{M_1}$ and $\mathcal{L}_{\text{seg}}^{M_2}$ denote the segmentation losses for modality $M_1$ and $M_2$ respectively. Similarly, the $\mathcal{L}_{\text{aux}}^{M_1}$ and $\mathcal{L}_{\text{aux}}^{M_2}$ indicate auxiliary prediction losses for two modalities.

| Methods | Cardiac CT | | | | | Cardiac MRI | | | | | Overall Mean |
|---|---|---|---|---|---|---|---|---|---|---|---|
| | LVM | LAC | LVC | AA | Mean | LVM | LAC | LVC | AA | Mean | |
| | | | | | | Dice Coefficient (avg.± std., %) ↑ | | | | | |
| Payer *et al.*[MMWHS18] | 87.2±3.9 | 92.4±3.6 | 92.4±3.3 | 91.1±18.4 | 90.8 | 75.2±12.1 | 81.1±13.8 | 87.7±7.7 | 76.6±13.8 | 80.2 | 85.5 |
| UMMKD[TMI20] | 88.5±3.1 | 91.5±3.1 | 93.1±2.1 | 93.6±4.3 | 91.7 | 80.8±3.0 | 86.5±6.5 | 93.6±1.8 | 83.1±5.8 | 86.0 | 88.8 |
| Backbone | 90.0±3.2 | 92.5±2.9 | 92.6±3.0 | 87.4±3.8 | 90.6 | 79.9±4.6 | 85.3±3.9 | 92.0±2.7 | 84.9±2.9 | 85.5 | 88.1 |
| Baseline | 90.6±2.8 | 92.6±2.8 | 93.2±2.5 | 88.9±3.4 | 91.3 | 80.9±4.0 | 86.3±3.8 | 92.9±2.3 | 85.8±3.5 | 86.5 | 88.9 |
| Joint Training | 89.1±2.8 | 93.0±2.7 | 92.8±3.3 | 91.2±2.6 | 91.5 (+0.2) | 80.2±3.9 | 86.5±4.5 | 92.0±3.0 | 86.1±3.8 | 86.2 (-0.3) | 88.9 (+0.0) |
| Our(w/o CR) | 90.0±2.3 | 93.8±2.1 | 93.4±2.4 | 94.0±2.0 | 92.8 (+1.5) | 81.0±3.1 | 87.4±3.6 | 93.5±2.1 | 87.8±3.0 | 87.4 (+0.9) | 90.1 (+1.2) |
| **Ours(w/ CR)** | **90.9±2.0** | **94.8±1.6** | **94.5±2.1** | **95.9±1.4** | **94.0** (+2.7) | **81.6±2.5** | **89.6±3.3** | **94.4±1.3** | **89.2±2.8** | **88.7** (+2.2) | **91.4** (+2.5) |
| | | | | | | Average Symmetric Surface Distance (avg.± std., *mm*) ↓ | | | | | |
| Payer *et al.*[MMWHS18] | - | - | - | - | - | - | - | - | - | - | |
| UMMKD[TMI20] | - | - | - | - | - | - | - | - | - | - | |
| Backbone | 1.67±0.46 | 1.95±0.54 | 1.43±0.47 | 1.51±0.41 | 1.64 | 2.12±1.57 | 1.74±0.85 | 1.41±0.81 | 3.74±1.68 | 2.25 | 1.95 |
| Baseline | 1.49±0.33 | 1.84±0.44 | 1.38±0.35 | 1.46±0.28 | 1.54 | 1.71±1.43 | 1.37±0.64 | 1.46±0.89 | 2.69±1.27 | 1.86 | 1.70 |
| Joint Training | 1.58±0.35 | 1.70±0.44 | 1.39±0.35 | 1.33±0.38 | 1.50 (-0.04) | 1.87±0.92 | 1.47±0.40 | 1.42±0.55 | 3.13±1.41 | 1.97 (+0.11) | 1.74 (+0.04) |
| Our(w/o CR) | 1.34±0.31 | 1.63±0.46 | 1.32±0.27 | 1.10±0.29 | 1.35 (-0.19) | 1.84±0.81 | **1.22±0.53** | 1.39±0.58 | 2.05±1.10 | 1.63 (-0.23) | 1.49 (-0.21) |
| **Ours(w/ CR)** | **1.31±0.27** | **1.49±0.38** | **1.22±0.27** | **1.00±0.24** | **1.26** (-0.28) | **1.55±0.78** | 1.24±0.34 | **1.27±0.32** | **2.01±0.95** | **1.52** (-0.34) | **1.39** (-0.31) |

Table 1: The performance of cardiac substructure segmentation by using 2D Transformer.

## 3. Experiments

We evaluate the performance of our method on the two multi-modality segmentation tasks, *i.e.* Cardiac Substructure Segmentation and Abdominal Multi-organ Segmentation, under 2D and 3D model configurations respectively. For fairness, we provide four experimental settings: (1) Backbone that is separately trained with single modality; (2) Baseline that adds auxiliary prediction loss $\mathcal{L}_{aux}$ based on Backbone; (3) Joint Training that shares the entire backbone model to jointly train multiple modalities; (4) Ours (w/o CR) that introduces the modality-aware channel-wise multiplication mechanism in each transformer block of the shared encoder and decoder, as illustrated in **Appendix** B.1; (5) Ours (w/ CR) that is our full cross-modal learning strategy by adding two types of consistency terms $\mathcal{L}_{mcr}$, $\mathcal{L}_{icr}$. We assess segmentation performance using the Volume Dice Coefficient (Dice, %) and Average Symmetric Surface Distance (ASD, *mm*) metrics.

### 3.1. Cardiac Substructure Segmentation

**2D Configuration.** We employ the Multi-Modality Whole Heart Segmentation Challenge 2017 dataset (Zhuang et al., 2019) to perform multi-class cardiac structure segmentation. We adopt a 2D U-shaped Transformer named Swin-Unet (Cao et al., 2021) as the backbone. Please refer to the **Appendix** C for more details about the dataset and network.

**Main Results.** Based on the 2D Backbone, we extend our Baseline model by the auxiliary prediction loss $\mathcal{L}_{aux}$ in Eqn. 9, which achieves average segmentation Dice of 91.3% on CT and 86.5% on MRI and outperforms the Backbone model, as well as the MICCAI-MMWHS challenge winner Payer *et al.* (Payer et al., 2017) that also deploys single modality training. This demonstrates that similar to a form of deep supervision, calibrating multi-scale semantic-aware feature maps improves final segmentation performance noticeably.

For multi-modality training, we share the entire backbone model (*i.e.*, encoder, decoder, and prediction head) for multi-modality training, denoted as Joint Training. However, there is a decrease in segmentation results, *i.e.* Dice of 91.5% on CT and 86.2% on MRI. This suggests that in such a situation, modality discrepancy has a significant impact on learned feature representations. We then introduce a modality-aware channel-wise multiplication mechanism into each Transformer block of shared encoder and decoder based on the Joint Training model, as in **Appendix** B.1. This training scheme denoted as Ours (w/o CR) further improves segmentation results to 92.8% on CT and

| Methods | Abdominal CT | | | | | Abdominal MRI | | | | | Overall Mean |
|---|---|---|---|---|---|---|---|---|---|---|---|
| | Liver | Spleen | R-kdy | L-kdy | Mean | Liver | Spleen | R-kdy | L-kdy | Mean | |
| Dice Coefficient (avg.± std., %) ↑ | | | | | | | | | | | |
| UMMKD[TMI20] | 92.7±1.8 | 93.7±1.7 | **94.0±0.7** | 89.5±3.9 | 92.4 | 90.3±2.8 | 87.4±1.1 | 91.0±1.5 | 88.3±1.7 | 89.3 | 90.8 |
| Backbone | 93.2±2.8 | 90.9±2.4 | 86.9±2.7 | 87.5±3.8 | 89.5 | 91.7±4.0 | 87.2±3.2 | 90.9±2.7 | 90.6±3.3 | 90.1 | 89.9 |
| Baseline | 93.6±2.2 | 92.1±2.6 | 87.9±1.8 | 87.7±3.6 | 90.3 | 92.9±3.3 | 87.8±2.9 | 92.0±2.5 | 91.4±3.1 | 91.0 | 90.7 |
| Joint Training | 94.0±2.6 | 92.5±2.2 | 87.8±2.9 | 87.9±3.4 | 90.6 (+0.3) | 92.6±3.4 | 87.3±3.1 | 91.2±1.9 | 90.8±3.7 | 90.5 (-0.5) | 90.5 (-0.2) |
| Ours(w/o CR) | 94.6±2.2 | 93.3±1.8 | 88.9±2.3 | 88.7±3.5 | 91.4 (+1.1) | 93.8±2.3 | 88.5±2.7 | 92.7±1.5 | 91.5±3.1 | 91.6 (+0.6) | 91.5 (+0.8) |
| **Ours(w CR)** | **95.8±1.4** | **94.9±1.3** | 92.3±0.9 | **91.8±2.2** | **93.7** (+3.4) | **94.7±1.5** | **89.9±1.2** | **93.6±0.8** | **93.0±1.4** | **92.8** (+1.8) | **93.3** (+2.6) |
| Average Symmetric Surface Distance (avg.± std., mm) ↓ | | | | | | | | | | | |
| UMMKD[TMI20] | - | - | - | - | - | - | - | - | - | - | - |
| Backbone | 1.19±0.91 | 1.18±0.82 | 1.84±1.06 | 1.10±0.78 | 1.33 | 1.20±0.68 | 1.27±0.79 | 1.36±0.94 | 1.37±0.71 | 1.30 | 1.31 |
| Baseline | 1.12±0.75 | 0.98±0.68 | 1.60±0.93 | 1.05±0.65 | 1.19 | 1.07±0.52 | 1.19±0.76 | 1.22±0.80 | 1.23±0.64 | 1.18 | 1.18 |
| Joint Training | 1.03±0.62 | 0.85±0.51 | 1.87±0.84 | 0.96±0.58 | 1.18 (-0.01) | 1.19±0.56 | 1.32±0.73 | 1.27±0.85 | 1.34±0.67 | 1.28 (+0.10) | 1.23 (+0.05) |
| Ours(w/o CR) | 0.94±0.58 | 0.75±0.37 | 1.37±0.61 | 0.82±0.43 | 0.97 (-0.22) | 1.01±0.49 | 1.18±0.64 | 1.03±0.69 | 1.15±0.53 | 1.09 (-0.09) | 1.03 (-0.15) |
| **Ours(w CR)** | **0.87±0.29** | **0.58±0.17** | **0.84±0.32** | **0.72±0.24** | **0.75** (-0.44) | **0.83±0.36** | **0.56±0.23** | **0.85±0.39** | **0.83±0.37** | **0.77** (-0.41) | **0.76** (-0.42) |

Table 2: The results of abdominal multi-organ segmentation by using 3D Transformer.

87.4% on MRI, demonstrating the efficiency of modality-specific activation calibration. Finally, Ours (w/o CR) with $\mathcal{L}_{\mathrm{mcr}}$ and $\mathcal{L}_{\mathrm{icr}}$ marked as Ours (w/ CR) in Table 5 achieves an overall Dice of 91.4% (i.e. the average of 94.0% on CT and 88.7% on MRI). When compared to the current state-of-the-art multi-modal approach UMMKD (Dou et al., 2020), our segmentation result has a 2.6% promotion on overall mean Dice. In addition, our method achieves the lowest overall mean ASD (i.e. 1.39mm) among all compared approaches. We also present a visual representation of the segmentation results for quantitative comparison in **Appendix** 4.

## 3.2. Abdominal Multi-organ Segmentation

**3D Configuration.** We adopt public CT data from (Landman et al., 2015) with 30 patients, and the T2-SPIR MRI data from the ISBI 2019 CHAOS Challenge (Kavur et al., 2021). We employ a 3D U-shaped Transformer named nnFormer (Zhou et al., 2021) as the backbone. The detailed dataset and network descriptions are in **Appendix** C

**Main Results.** Here, the Baseline model is the Backbone with $\mathcal{L}_{\mathrm{aux}}$, which is the same as the procedure in the 2D case. The Dice values raise to 90.3% on CT and 91.0% on MRI, which is shown in Table 6. Likewise, sharing the entire Backbone model in Joint Training causes a slight segmentation performance drop, i.e. Dice of 90.6% on CT and 90.5% on MRI. By integrating modality-specific activation, Ours (w/o CR) improves the Dice values to 91.4% on CT and 91.6% on MRI, outperforming Backbone, Baseline, and Joint Training by a large margin. Furthermore, by using the $\mathcal{L}_{\mathrm{mcr}}$ and $\mathcal{L}_{\mathrm{icr}}$, our full multi-modality learning scheme marked as Ours (w/ CR) achieves the best segmentation results of 93.3% overall mean Dice and 0.76mm overall mean ASD. Compared with the counterpart, our scheme also outperform UMMKD by a significant margin, i.e. the Dice value of 1.3% on CT and 3.5% on MRI. The visual segmentation results for quantitative comparison are in **Appendix** 4.

## 4. Conclusion

This paper studies how to train the single segmentation model to conduct unpaired multi-modal medical image predictions. A novel plug-and-play External Attention Module (EAM) is introduced to regulate the backbone network to obey the structured semantic consistency for different modalities, i.e. modality-specific class representations and image-specific inter-class correlations. In the test phase, the EAMs can be removed, maintaining the simplicity of the network. Extensive experimental results show the effectiveness of our method.

## Acknowledgments

The work is supported in part by National Key R&D Program of China under grant No. 2022ZD0116004, by the Young Scientists Fund of the National Natural Science Foundation of China under grant No. 62106154, by Natural Science Foundation of Guangdong Province, China (General Program) under grant No.2022A1515011524, by Guangdong Basic and Applied Basic Research Foundation under Grant No. 2017A030312006, by Shenzhen Science and Technology Program ZDSYS20211021111415025, and by the Guangdong Provincial Key Laboratory of Big Data Computing, The Chinese University of Hong Kong (Shenzhen).

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

## Appendix A. Related Work

### A.1. Domain adaptation.

In medical image analysis, severe domain shift has been a long-standing obstacle to knowledge transfer between unpaired modalities obtained by different physical principles of imaging. To address this problem, research works on domain adaptation of models either in a semi-supervised (Zhu et al., 2021) or unsupervised (Chen et al., 2020; Huo et al., 2018) manner, aiming to effectively improve cross-modality representation learning. To use data more effectively, some works also focus on full-supervised multi-modality learning, transferring the knowledge that can promote each other between different modalities. In (Valindria et al., 2018), Valindria *et al.* discussed the effectiveness of various dual-stream architectures, demonstrating that the domain shift between two unpaired modalities limits mutual information sharing. Following this, Dou *et al.* (Dou et al., 2020) proposed a multi-modal learning approach by employing modality-specific internal normalization layers to compute respective statistics of each modality, and conduct cross-modality knowledge distilling to reduce the gap of prediction distributions between modalities. Instead of carefully constructing multi-modal networks with different feature fusion strategies, our work aims to design the external plug-and-play module to help a single model to establish the structured semantic consistency of different modalities, realizing multi-modal predictions.

### A.2. Vision Transformer.

Currently, the studies about vision Transformer (Dosovitskiy et al., 2020) also achieve great progress in image analysis and understanding. ViT is a convolution-free network architecture, which directly employs the attention mechanism to capture the long-range dependence of a sequence of non-overlapping image patches, Followed by ViT, Touvron *et al.* (Touvron et al., 2021a) proposed DeiT that introduced a distillation strategy for Transformer to help with ViT training. And many other ViT variants are also proposed (Chen et al., 2021; Wang et al., 2021a; Rao et al., 2021; Zheng et al., 2021; Liu et al., 2021) which achieve promising performance compared with its counterpart CNNs on various vision tasks, such as image classification and semantic segmentation. Inspired by SwinTransformer, Cao *et al.* (Cao et al., 2021) proposed a pure U-shaped transformer named Swin-Unet, where the architecture utilizes SwinTransformer block as the basic unit to improve its capacity of feature representation for 2D medical image segmentation. Furthermore, to explore Transformer's ability to learn volumetric representations from 3D medical volumes, Zhou *et al.* (Zhou et al., 2021) proposed nnFormer to interleave convolution and self-attention for medical

volumetric segmentation, achieving tremendous progress over previous transformer-based medical segmentation methods.

## Appendix B. Methodology Details

### B.1. Transformer Block

Motivated by the LayerScale (Touvron et al., 2021b), we introduce modality-aware channel-wise multiplication on the output of each residual operation in the Transformer block, with the goal of calibrating modality-specific activation in the channel dimension to narrow the discrepancy between representations from different modalities. Given the modality-specific class embeddings $\mathbf{Q} \in \mathbb{R}^{Z \times 4C}$ of a certain modality, where $Z$ is the total number of classes and is constant across all modalities, and $4C$ denotes the channel dimension, we aggregate its semantic information by using linear projection to produce modality-specific channel weight $\mathbf{\Omega} \in \mathbb{R}^{4C}$. We then project $\mathbf{\Omega}$ to the corresponding feature dimensions (*e.g.,* from $4C$ to $D$ for the specific scale), and the diag($\cdot$) operation is adopted to generate diagonal matrix to calibrate modality-specific activation as,

$$\mathbf{\Omega} = \mathbf{w}_1 \mathbf{Q}, \ \ \mathbf{\Phi}_1 = \mathrm{diag}(\mathbf{\Omega}\mathbf{W}_2), \ \ \mathbf{\Phi}_2 = \mathrm{diag}(\mathbf{\Omega}\mathbf{W}_3), \tag{13}$$

$$\mathbf{X}'_l = \mathbf{X}_l + \mathbf{\Phi}_1 \circledast \mathrm{MSA}( \ \mathrm{Norm}(\mathbf{X}_l) \ ), \tag{14}$$

$$\mathbf{X}_{l+1} = \mathbf{X}'_l + \mathbf{\Phi}_2 \circledast \mathrm{FFN}( \ \mathrm{Norm}(\mathbf{X}'_l) \ ), \tag{15}$$

where $\mathbf{w}_1 \in \mathbb{R}^Z$, $\mathbf{W}_2 \in \mathbb{R}^{4C \times D}$, $\mathbf{W}_3 \in \mathbb{R}^{4C \times D}$, MSA($\cdot$) and FFN($\cdot$) denote the multi-head attention layer and the feed-forward layer respectively. Norm($\cdot$) indicates the LayerNorm operation, and $\circledast$ is the channel-wise multiplication. $\mathbf{X}_l$ and $\mathbf{X}_{l+1}$ denote the input and output of $(l+1)$-th transformer block. As shown in Fig. 1, we feed the modality-specific class embeddings into the Transformer decoder, and we implement the above scheme in all Transformer blocks to improve accuracy even further. **It is worth noting** that such modality-aware channel-wise recalibration is an optional operation if we require to drop the modality-aware query as well in inference.

### B.2. External Attention Module

In the main article, we design the External Attention Module (EAM) to update the learnable class embeddings from modality-specific to image-specific, and extract the semantic correlations of a specific image at multiple scales. Fig. 2 shows the details of EAM for the Stage-1 of Transformer Decoder, which adopts *modality-specific class embeddings* and feature maps at scale-1 as the input. In practice, we replace *modality-specific class embeddings* with *image-specific class embeddings* for Stage-2 and Stage-3, which is illustrated in Fig. 1.

## Appendix C. Implementation Details

### C.1. Datasets

**Cardiac Substructure Segmentation.** We employ the Multi-Modality Whole Heart Segmentation Challenge 2017 dataset (Zhuang et al., 2019) to perform multi-class cardiac structure segmentation. The dataset is composed of unpaired 20 CT and 20 MRI scans collected

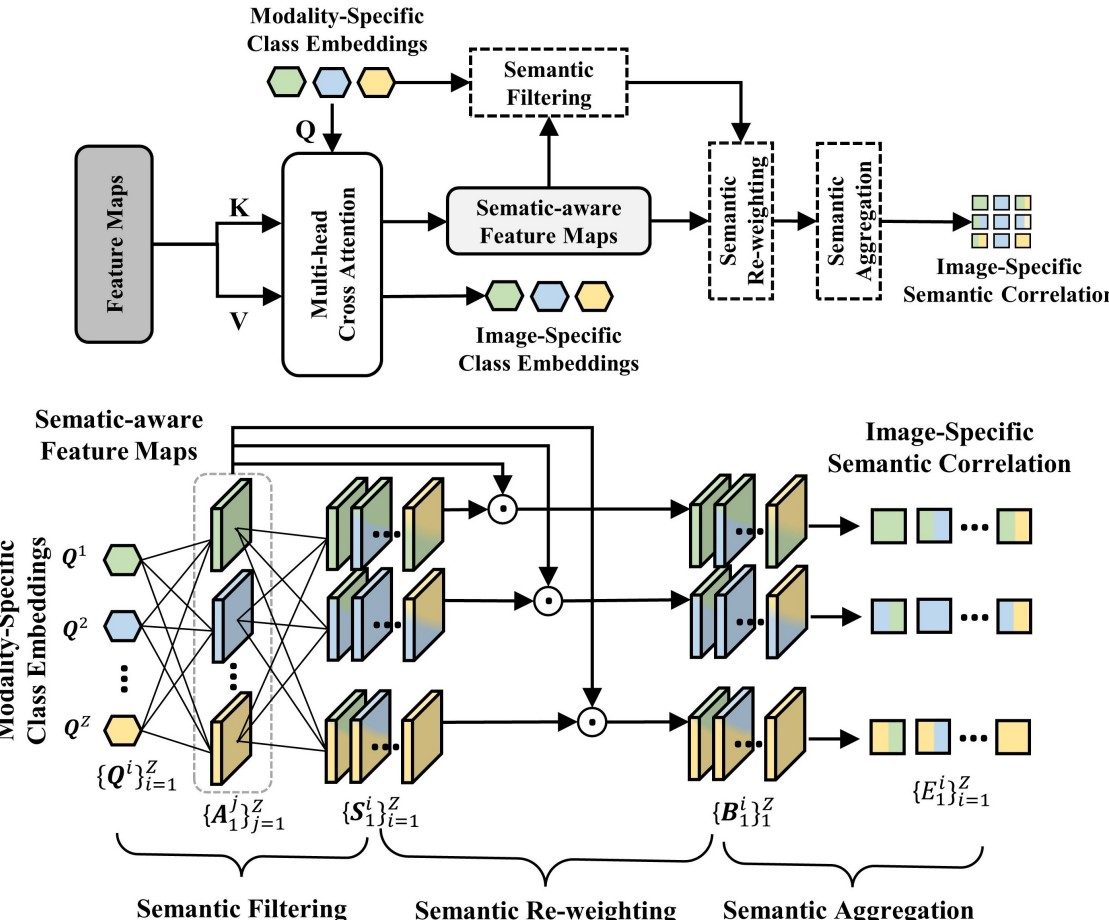

Figure 2: External Attention Module (EAM)

from various patients and sites. We intend to train the segmentation network to recognize four organs for each modality: `left ventricle myocardium` (LVM), `left atrium blood cavity` (LAC), `left ventricle blood cavity` (LVC), and `ascending aorta` (AA). Since the UMMKD approach (Dou et al., 2020) achieves state-of-the-art performance, we take the same partitioning (*i.e.* training, validation, and test) and pre-processing of the dataset as the UMMKD reported for a fair comparison. Specifically, we align with them to split each modality into 70% for training, 10% for validation, and 20% for testing. Before feeding into the network, we first resample both modalities to isotropic $1\text{mm}^3$ and crop their central heart region by a 3D bounding box which has a fixed coronal plane size of $256 \times 256$. For each 3D cropped image, we remove the top 2% of its intensity histogram to reduce artifacts, which is then normalized to zero mean and unit variance.

**Abdominal Multi-organ Segmentation.** We adopt public CT data from (Landman et al., 2015) with 30 patients, and the T2-SPIR MRI data from the ISBI 2019 CHAOS Challenge (Kavur et al., 2021) with 20 volumes. We focus on adapting the segmentation network to delineate four abdominal organs: `liver`, `right kidney` (R-kdy), `left kidney` (L-kdy), and `spleen`. Since the UMMKD method does not report the detailed partitioning

| Layer | 2D Transformer (Cao et al., 2021) | 3D Transformer (Zhou et al., 2021) |
|---|---|---|
| Transformer Block | Swin Transformer Block (Liu et al., 2021) | Volume-based Self-attention |
| Down-sampling | Patch Merging | Strided Convolution |
| Up-sampling | Patch Expanding | Strided Convolution |
| $[G_1, G_2, G_3, G_4]$ | $[2, 2, 2, 2]$ | $[2, 2, 2, 2]$ |
| $[L_1, L_2, L_3, L_4]$ | $[2, 2, 2, 2]$ | $[2, 2, 2, 2]$ |
| Patch Size | $4 \times 4$ | $4 \times 4 \times 2$ |

Table 3: Detailed configurations of 2D Transformer and 3D Transformer architectures corresponding to Fig. 1, where $\{G_i\}_{i=1}^4$ and $\{L_i\}_{i=1}^4$ denote the basic number of Transformer blocks for the Encoder and Decoder respectively.

of these two datasets, we attempt to align their practices for a more fair comparison. The UMMKD method (Dou et al., 2020), in particular, employs only 9 MRI scans from the MRI dataset and removes a low-quality CT scan from the CT dataset. As a result, we randomly select 9 MRI cases from all 20 MRI scans five times and use the entire CT images from the CT dataset to demonstrate the robustness of our method and then average the results of the five randomized experiments to compare with their method. In accordance with UMMKD, we randomly divided each data modality into 70% for training, 10% for validation, and 20% for testing. Following UMMKD's pre-processing methods, we first resample them into around $1.5 \times 1.5 \times 8.0$ mm$^3$ with a size of $256 \times 256$ in the coronal plane to eliminate the huge difference in voxel-spacing between two datasets, and then perform intensity normalization to zero mean and unit variance for each modality.

## C.2. Evaluation Metrics

We assess segmentation performance using the Volume Dice Coefficient (Dice, %) and Average Symmetric Surface Distance (ASD, $mm$) metrics, calculating the average and standard deviation of segmentation results for each class (Dou et al., 2020). Note that the final segmentation results (mean and standard variance) in the abdominal multi-organ segmentation task are calculated from the results of all the test samples in the five random tests.

## C.3. Model Configuration

As in the main article, we implement 2D and 3D Transformers with various network architectures as the backbone to demonstrate the flexibility and general efficacy of our method. Specifically, we adopt a 2D Transformer-based U-shaped Encoder-Decoder architecture named Swin-Unet (Cao et al., 2021) for the multi-class cardiac structure segmentation. For the abdominal multi-organ segmentation, we employ a 3D U-shaped Transformer with a volume-based self-attention mechanism and strided convolution named nnFormer (Zhou et al., 2021). We report the detailed network architecture in Table. 3.

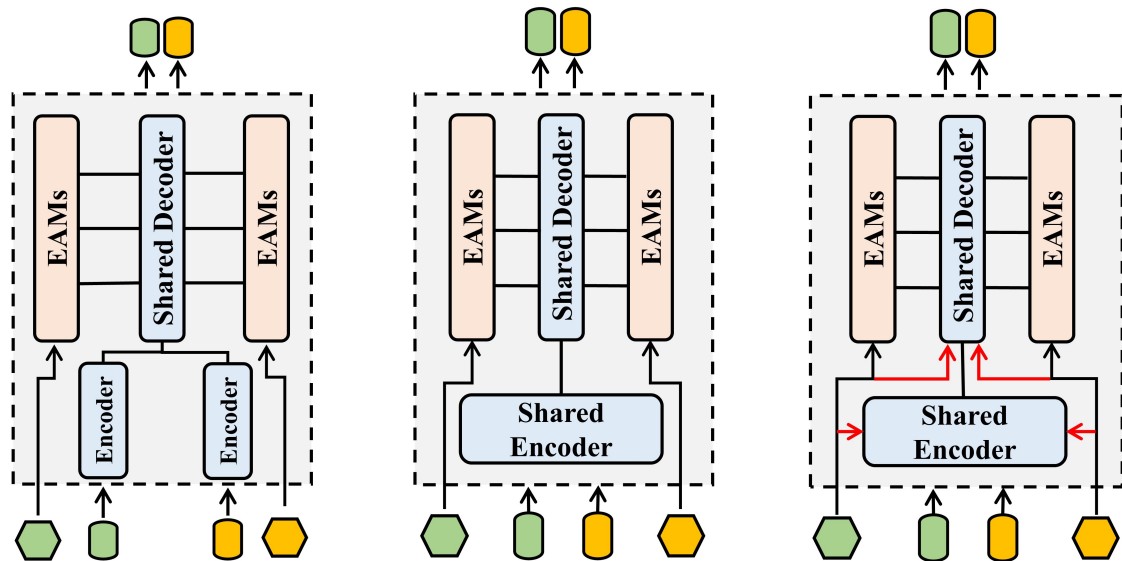

Figure 3: Three versions of Joint architecture: (a) Using modality-specific Encoder and EAMs while sharing Decoder; (b) Sharing both Encoder and Decoder while keeping EAMs as modality-specific; (c) Incorporating the modality-aware channel-wise multiplication mechanism into each transformer block of shared Encoder and Decoder in the architecture (b). Note that the input hexagons and rectangles denote modality-specific class embeddings and patch tokens, respectively.

## Appendix D. Few-shot Domain Adaptation

### D.1. Experiment Setting

To assess the effectiveness of our method when training with limited samples of one modality, we randomly select 1 or 3 samples from one modality's (CT or MRI) training set while training the model with all training data from another modality. We conduct all of the few-shot domain adaptation experiments on MICCAI-MMWHS dataset.

### D.2. Main Results

As shown in Table 4, when our multi-modal learning framework is trained with only 1 annotated MRI image and all CT images, the value of mean Dice for MRI is only slightly lower than Baseline (*i.e.* 0.4%), which is trained with all MRI images alone. This demonstrates that when using complete CT training data, the proposed method can eliminate the impact of insufficient MRI training samples. In such a case, the mean Dice of CT improves to 92.7% and the mean ASD decreases to 1.41mm, evidencing that the feature representation of CT could also be enhanced by using only 1 MRI image. Moreover, when the number of MRI training samples increases to 3, the mean Dice values improve to 93.3% on CT and 86.9% on MRI, surpassing the Baseline model.

When training with only 1 CT image and all of MRI images using our scheme, the mean Dice of CT drops by 0.7% compared with Baseline model which is only trained with all of CT images. Also, giving 1 annotated CT image improves the mean Dice of MRI by

| Methods | Cardiac CT | | | | | Cardiac MRI | | | | | Overall Mean |
|---|---|---|---|---|---|---|---|---|---|---|---|
| | LVM | LAC | LVC | AA | Mean | LVM | LAC | LVC | AA | Mean | |
| Dice Coefficient (avg.± std., %) ↑ | | | | | | | | | | | |
| Baseline | 90.6±2.8 | 92.6±2.8 | 93.2±2.5 | 88.9±3.4 | 91.3 | 80.9±4.0 | 86.3±3.8 | 92.9±2.3 | 85.8±3.5 | 86.5 | 88.9 |
| Ours (w/ CR) | 90.9±2.0 | 94.8±1.6 | 94.5±2.1 | 95.9±1.4 | 94.0 | 81.6±2.5 | 89.6±3.3 | 94.4±1.3 | 89.2±2.8 | 88.7 | 91.4 |
| One annotated MRI | 90.7±2.7 | 93.1±2.5 | 93.3±2.8 | 93.7±2.4 | 92.7 (+1.4) | 78.8±5.8 | 85.2±4.6 | 92.5±3.2 | 87.9±4.1 | 86.1 (-0.4) | 89.4 (+0.5) |
| Three annotated MRI | 90.4±2.3 | 93.7±2.0 | 93.9±2.7 | 95.0±1.9 | 93.3 (+1.2) | 80.6±4.7 | 86.0±4.1 | 92.6±2.9 | 88.2±3.3 | 86.9 (+0.4) | 90.1 (+1.4) |
| One annotated CT | 88.2±3.4 | 91.2±3.5 | 91.4±2.2 | 91.6±3.3 | 90.6 (-0.7) | 80.8±4.2 | 87.9±4.3 | 93.1±2.1 | 88.6±2.9 | 87.6 (+1.1) | 89.1 (+0.2) |
| Three annotated CT | 88.9±3.1 | 91.7±2.9 | 92.4±2.3 | 92.8±2.8 | 91.5 (+0.2) | 81.1±3.7 | 88.4±3.5 | 93.5±1.8 | 88.9±2.7 | 88.0 (+1.5) | 89.7 (+0.8) |
| Average Symmetric Surface Distance (avg.± std., mm) ↓ | | | | | | | | | | | |
| Baseline | 1.49±0.33 | 1.84±0.44 | 1.38±0.35 | 1.46±0.28 | 1.54 | 1.71±1.43 | 1.37±0.64 | 1.46±0.89 | 2.69±1.27 | 1.86 | 1.70 |
| Ours (w/ CR) | 1.31±0.27 | 1.49±0.38 | 1.22±0.27 | 1.00±0.24 | 1.26 | 1.55±0.78 | 1.24±0.34 | 1.27±0.32 | 2.01±0.95 | 1.52 | 1.39 |
| One annotated MRI | 1.34±0.34 | 1.68±0.48 | 1.37±0.36 | 1.25±0.30 | 1.41 (-0.13) | 1.89±1.59 | 2.04±0.93 | 1.63±1.01 | 2.36±2.18 | 1.98 (+0.12) | 1.70 (+0.00) |
| Three annotated MRI | 1.41±0.37 | 1.55±0.41 | 1.34±0.47 | 1.16±0.26 | 1.37 (-0.17) | 1.73±1.28 | 1.82±0.87 | 1.51±0.83 | 2.24±1.06 | 1.83 (-0.04) | 1.60 (-1.00) |
| One annotated CT | 1.86±0.58 | 2.13±0.86 | 1.92±0.92 | 1.31±0.35 | 1.81 (+0.27) | 1.75±1.31 | 1.44±0.47 | 1.42±0.75 | 2.08±1.02 | 1.67 (-0.19) | 1.74 (+0.04) |
| Three annotated CT | 1.69±0.44 | 1.95±0.71 | 1.61±0.59 | 1.26±0.37 | 1.63 (+0.09) | 1.62±0.94 | 1.34±0.52 | 1.36±0.49 | 2.14±1.12 | 1.62 (-0.24) | 1.62 (-0.08) |

Table 4: The results of **few-shot domain adaptation** on cardiac segmentation by using 2D network.

1.1% and decreases the mean ASD of MRI by 0.17mm, highlighting the mutual promotion between CT and MRI. Further, using 3 CT images raises the mean Dice to 91.5% on CT and 88.0% on MRI, defeating UMMKD by 2.0% on MRI while maintaining competitive performance on CT.

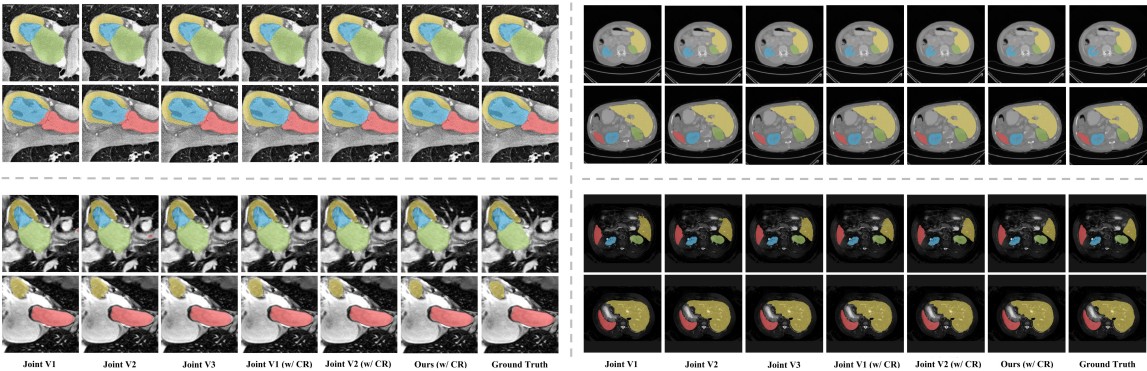

Figure 4: Visualization results on cardiac segmentation task and abdominal multi-organ task. **Left:** AA, LAC, LVC and LVM. **Right:** Spleen, R-kdy, L-kdy and Liver. The corresponding colormaps are red, green, blue and yellow, respectively.

## Appendix E. Ablation Studies

### E.1. Three Versions of Joint Architecture

To thoroughly evaluate the effectiveness of the different joint architectures and two consistency regularization terms of our method, we provide three degenerate models of the proposed method with or without the two consistency regularization losses. The detailed architectures are shown in Fig. 3. We report the six experimental settings that conduct the fixed 2D or 3D Transformer and hyper-parameters: (1) **Joint V1** that is a joint Transformer with modality-specific encoder and EAMs, shared decoder, which is a conventional multi-modal learning architecture (Nie et al., 2016) as shown in Fig. 3-1; 2) **Joint V2** (de-

noted as **Joint Training** in the main article) is a joint Transformer with shared encoder and decoder, and modality-specific EAMs, as shown in Fig. 3-2; 3) **Joint V3** (denoted as **Ours (w/o CR)** in the main article) is trained based on Joint V2, implementing the modality-aware channel-wise multiplication mechanism in each transformer block of the shared encoder and decoder, as shown in Fig. 3-3; 4) **Joint V1 (w/ CR)** is trained based on Joint V1 Transformer with two types of consistency terms $\mathcal{L}_{\mathrm{mcr}}$, $\mathcal{L}_{\mathrm{icr}}$; 5) **Joint V2 (w/ CR)** is trained based on Joint V2 Transformer with two types of consistency terms $\mathcal{L}_{\mathrm{mcr}}$, $\mathcal{L}_{\mathrm{icr}}$; 6) **Ours (w/ CR)** that is our full cross-modal learning strategy by adding two types of consistency terms $\mathcal{L}_{\mathrm{mcr}}$, $\mathcal{L}_{\mathrm{icr}}$.

### E.1.1. CARDIAC SUBSTRUCTURE SEGMENTATION.

As shown in Fig. 3-1, based on the Baseline model, Joint V1 exemplifies a conventional multi-modal Transformer architecture by using modality-specific encoders and EAMs while sharing decoders. This model raises the Dice value to 91.9% for CT and 86.8% for MRI. This clearly illustrates the Transformer's ability to deal with multi-modal data with large appearance differences at the same time, as well as the potential for mutual promotion between different modalities by sharing some modules. In Fig. 3-2, Joint V2 targets to further share decoder between different modalities to improve the parameter efficiency of multi-modal Transformer. However, there is a decrease in segmentation results, *i.e.* Dice of 91.5% on CT and 86.2% on MRI. This suggests that in such a situation, modality discrepancy has a significant impact on learned feature representations without introducing any semantic consistency constraints. In Fig. 3-3, we introduce a modality-aware channel-wise multiplication mechanism into each Transformer block of shared encoder and decoder. This scheme improves segmentation results to 92.8% on CT and 87.4% on MRI, demonstrating the efficiency of modality-specific activation calibration.

By leveraging proposed two types of consistency regularization terms, *i.e.* $\mathcal{L}_{\mathrm{mcr}}$ and $\mathcal{L}_{\mathrm{icr}}$, the three multi-modal Transformer architectures Joint V1, Joint V2 and Joint V3 are all boosted. Specifically, The Joint V1 (w/ CR) and Joint V2 (w/ CR) improve the Dice value by 1.9% and 1.7% on CT respectively. The Joint V3 with $\mathcal{L}_{\mathrm{mcr}}$ and $\mathcal{L}_{\mathrm{icr}}$ is marked as **Ours (w/ CR)** in Table 5, and it achieves an overall Dice of 91.4% (*i.e.* the average of 94.0% on CT and 88.7% on MRI). In Fig. 4 (Left), we also present a visual representation of the segmentation results for quantitative comparison.

### E.1.2. ABDOMINAL MULTI-ORGAN SEGMENTATION.

Likewise, the multi-modal Transformer architecture Joint V1 improves the Dice value by 0.4% on CT and 0.2% on MRI when compared to the Baseline model trained from a single modality. In contrast to Joint V1, sharing decoder in Joint V2 causes a slight segmentation performance drop when the number of parameters is reduced. By integrating modality-specific activation, Joint V3 improves the Dice values to 91.4% on CT and 91.6% on MRI, outperforming both Joint V2 and Joint V1 by a significant margin. Furthermore, by using the $\mathcal{L}_{\mathrm{mcr}}$ and $\mathcal{L}_{\mathrm{icr}}$, all three models improve significantly, and our full scheme achieves the best segmentation results of 93.3% overall mean Dice and 0.76mm overall mean ASD. Finally, we exhibit the visual segmentation results for quantitative comparison, as shown in Fig. 4 (Right).

Yang Zhu Wang Li Zhang

| Methods | Cardiac CT | | | | | Cardiac MRI | | | | | Overall Mean |
|---|---|---|---|---|---|---|---|---|---|---|---|
| | LVM | LAC | LVC | AA | Mean | LVM | LAC | LVC | AA | Mean | |
| | | | | | Dice Coefficient (avg.± std., %) ↑ | | | | | | |
| Backbone | 90.0±3.2 | 92.5±2.9 | 92.6±3.0 | 87.4±3.8 | 90.6 | 79.9±4.6 | 85.3±3.9 | 92.0±2.7 | 84.9±2.9 | 85.5 | 88.1 |
| Baseline | 90.6±2.8 | 92.6±2.8 | 93.2±2.5 | 88.9±3.4 | 91.3 | 80.9±4.0 | 86.3±3.8 | 92.9±2.3 | 85.8±3.5 | 86.5 | 88.9 |
| Joint V1 | 89.4±2.7 | 93.3±3.0 | 92.7±2.9 | 92.2±2.5 | 91.9 (+0.6) | 80.5±4.2 | 87.3±4.3 | 92.2±2.4 | 87.0±3.2 | 86.8 (+0.3) | 89.3 (+0.4) |
| Joint V2 | 89.1±2.8 | 93.0±2.7 | 92.8±3.3 | 91.2±2.6 | 91.5 (+0.2) | 80.2±3.9 | 86.5±4.5 | 92.0±3.0 | 86.1±3.8 | 86.2 (-0.3) | 88.9 (+0.0) |
| Joint V3 | 90.0±2.3 | 93.8±2.1 | 93.4±2.4 | 94.0±2.0 | 92.8 (+1.5) | 81.0±3.1 | 87.4±3.6 | 93.5±2.1 | 87.8±3.0 | 87.4 (+0.9) | 90.1 (+1.2) |
| Joint V1 (w/ CR) | 90.2±2.0 | 93.7±1.8 | 93.6±2.2 | 95.1±1.6 | 93.2 (+1.9) | **81.8±2.7** | 88.4±4.0 | 93.3±1.5 | 88.6±2.2 | 88.0 (+1.5) | 90.6 (+1.7) |
| Joint V2 (w/ CR) | 90.5±2.1 | 93.2±2.2 | 93.8±2.7 | 94.4±1.7 | 93.0 (+1.7) | 81.5±3.4 | 88.0±4.2 | 93.4±1.9 | 88.1±3.4 | 87.8 (+1.3) | 90.4 (+1.5) |
| **Ours (w/ CR)** | **90.9±2.0** | **94.8±1.6** | **94.5±2.1** | **95.9±1.4** | **94.0** (+2.7) | 81.6±2.5 | **89.6±3.3** | **94.4±1.3** | **89.2±2.8** | **88.7** (+2.2) | **91.4** (+2.5) |
| | | | | | Average Symmetric Surface Distance (avg.± std., mm) ↓ | | | | | | |
| Backbone | 1.67±0.46 | 1.95±0.54 | 1.43±0.47 | 1.51±0.41 | 1.64 | 2.12±1.57 | 1.74±0.85 | 1.41±0.81 | 3.74±1.68 | 2.25 | 1.95 |
| Baseline | 1.49±0.33 | 1.84±0.44 | 1.38±0.35 | 1.46±0.28 | 1.54 | 1.71±1.43 | 1.37±0.64 | 1.46±0.89 | 2.69±1.27 | 1.86 | 1.70 |
| Joint V1 | 1.63±0.38 | 1.64±0.40 | 1.47±0.32 | 1.19±0.27 | 1.48 (-0.06) | 1.99±1.07 | 1.36±0.57 | 1.51±0.73 | 2.89±1.33 | 1.94 (+0.08) | 1.71 (+0.01) |
| Joint V2 | 1.58±0.35 | 1.70±0.44 | 1.39±0.35 | 1.33±0.38 | 1.50 (-0.04) | 1.87±0.92 | 1.47±0.40 | 1.42±0.55 | 3.13±1.41 | 1.97 (+0.11) | 1.74 (+0.04) |
| Joint V3 | 1.34±0.31 | 1.63±0.46 | 1.32±0.27 | 1.10±0.29 | 1.35 (-0.19) | 1.84±0.81 | 1.22±0.53 | 1.39±0.58 | 2.05±1.10 | 1.63 (-0.23) | 1.49 (-0.21) |
| Joint V1 (w/ CR) | 1.29±0.26 | 1.65±0.44 | 1.28±0.30 | 1.13±0.28 | 1.34 (-0.20) | 1.76±0.87 | **1.15±0.36** | 1.48±0.52 | **1.92±1.18** | 1.58 (-0.28) | 1.46 (-0.24) |
| Joint V2 (w/ CR) | **1.27±0.28** | 1.73±0.47 | 1.34±0.32 | 1.08±0.25 | 1.36 (-0.18) | 1.69±0.85 | 1.18±0.47 | 1.50±0.64 | 1.97±1.26 | 1.59 (-0.27) | 1.47 (-0.23) |
| **Ours (w/ CR)** | 1.31±0.27 | **1.49±0.38** | **1.22±0.27** | **1.00±0.24** | **1.26** (-0.28) | **1.55±0.78** | 1.24±0.34 | **1.27±0.32** | 2.01±0.95 | **1.52** (-0.34) | **1.39** (-0.31) |

Table 5: The performance of cardiac substructure segmentation by using 2D Transformer.

## E.2. Effectiveness of Each Key Component

We employ four settings to verify the contribution of various key components: (a) we train the Joint V3 model without using any consistency regularization terms for both CT and MRI; (b) we only add the modality-level consistency regularization $\mathcal{L}_{\mathrm{mcr}}$ onto Joint V3, which corresponds to Eqn.10.; (c) we only add the instance-level consistency regularization $\mathcal{L}_{\mathrm{icr}}$ onto Joint V3, which corresponds to Eqn.11.; (d) we add both $\mathcal{L}_{\mathrm{mcr}}$ and $\mathcal{L}_{\mathrm{icr}}$ to accomplish our multi-modal learning scheme.

Table 7 reports the mean value of Dice and ASD for each class. Adding $\mathcal{L}_{\mathrm{mcr}}$ to Joint V3 improves the average Dice to 92.9% on CT and 92.2% on MRI and decreases the average ASD to 0.83mm on CT and 0.94mm on MRI. We also observe that the segmentation performance of each class improves significantly, regardless of CT or MRI, proving that aligning the representations of each class across various modalities could narrow the modalities' discrepancies in data distribution, allowing the network to be more generalized for both types of data. Furthermore, only adding $\mathcal{L}_{\mathrm{icr}}$ onto Joint V3 results in a mean Dice improvement of 1.8% on CT and 0.7% on MRI. This demonstrates that facilitating the network to dynamically learn the consistency of inter-class relationships at the image level within different modalities could also enhance the network's generalization to different modalities. Finally, by including both $\mathcal{L}_{\mathrm{mcr}}$ and $\mathcal{L}_{\mathrm{icr}}$, the Dice value further improves to 93.7% on CT and 92.8% on MRI, outperforming the variants that only add $\mathcal{L}_{\mathrm{mcr}}$ and $\mathcal{L}_{\mathrm{icr}}$, verifying that the two regularization terms can be used jointly to pursue the structured semantic consistency and effectively improve the segmentation performance.

## E.3. Setting of Temperature Hyper-parameters

The hyperparameter $\tau$ in $\mathcal{L}_{\mathrm{icr}}$ is used to control the softness of the inter-class probability distributions. We vary the value of $\tau$ to see how it affects the final segmentation results. As $\tau$ only influences the $\mathcal{L}_{\mathrm{icr}}$, we implement the Joint V3 model with only $\mathcal{L}_{\mathrm{icr}}$ added. Fig. 5 presents the segmentation performance changes (*i.e.* overall mean Dice). As the temperature $\tau$ increases from 1 to 4, the inter-class relation retains richer semantic information to guide the cross-modality learning. However, when $\tau > 4$, the more difficult pixel-wise constraints are exploited, and segmentation performance begins to deteriorate, possibly due

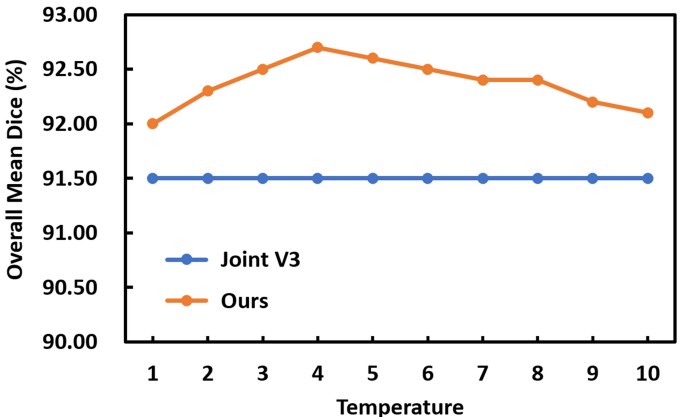

Figure 5: The effect of temperature $\tau$. Joint V3 is only used for comparison.

| Methods | Abdominal CT | | | | | Abdominal MRI | | | | | Overall Mean |
|---|---|---|---|---|---|---|---|---|---|---|---|
| | Liver | Spleen | R-kdy | L-kdy | Mean | Liver | Spleen | R-kdy | L-kdy | Mean | |
| Dice Coefficient (avg.± std., %) ↑ | | | | | | | | | | | |
| Backbone | 93.2±2.8 | 90.9±2.4 | 86.9±2.7 | 87.5±3.8 | 89.5 | 91.7±4.0 | 87.2±3.2 | 90.9±2.7 | 90.6±3.3 | 90.1 | 89.9 |
| Baseline | 93.6±2.2 | 92.1±2.6 | 87.9±1.8 | 87.7±3.6 | 90.3 | 92.9±3.3 | 87.8±2.9 | 92.0±2.5 | 91.4±3.1 | 91.0 | 90.7 |
| Joint V1 | 93.9±2.5 | 92.7±2.3 | 88.2±2.0 | 88.1±3.2 | 90.7 (+0.4) | 93.2±2.8 | 88.1±2.5 | 92.4±2.3 | 91.1±3.3 | 91.2 (+0.2) | 91.0 (+0.3) |
| Joint V2 | 94.0±2.6 | 92.5±2.2 | 87.8±2.9 | 87.9±3.4 | 90.6 (+0.3) | 92.6±3.4 | 87.3±3.1 | 91.2±1.9 | 90.8±3.7 | 90.5 (-0.5) | 90.5 (-0.2) |
| Joint V3 | 94.6±2.2 | 93.3±1.8 | 88.9±2.3 | 88.7±3.5 | 91.4 (+1.1) | 93.8±2.3 | 88.5±2.7 | 92.7±1.5 | 91.5±3.1 | 91.6 (+0.6) | 91.5 (+0.8) |
| Joint V1 (w/ CR) | 95.3±1.8 | 94.3±1.1 | 91.6±1.0 | 91.2±2.5 | 93.1 (+2.8) | 94.2±1.7 | 89.3±2.1 | 93.3±1.1 | 92.6±2.3 | 92.4 (+1.4) | 92.7 (+2.0) |
| Joint V2 (w/ CR) | 95.1±1.7 | 93.9±1.5 | 91.3±1.4 | 91.0±2.7 | 92.8 (+2.5) | 94.0±1.9 | 89.1±1.8 | 92.8±0.6 | 92.2±2.0 | 92.0 (+1.0) | 92.4 (+1.7) |
| **Ours (w/ CR)** | **95.8±1.4** | **94.9±1.3** | **92.3±0.9** | **91.8±2.2** | **93.7** (+3.4) | **94.7±1.5** | **89.9±1.2** | **93.6±0.8** | **93.0±1.4** | **92.8** (+1.8) | **93.3** (+2.6) |
| Average Symmetric Surface Distance (avg.± std., mm) ↓ | | | | | | | | | | | |
| Backbone | 1.19±0.91 | 1.18±0.82 | 1.84±1.06 | 1.10±0.78 | 1.33 | 1.20±0.68 | 1.27±0.79 | 1.36±0.94 | 1.37±0.71 | 1.30 | 1.31 |
| Baseline | 1.12±0.75 | 0.98±0.68 | 1.60±0.93 | 1.05±0.65 | 1.19 | 1.07±0.52 | 1.19±0.76 | 1.22±0.80 | 1.23±0.64 | 1.18 | 1.18 |
| Joint V1 | 1.07±0.56 | 0.79±0.45 | 1.45±0.71 | 0.93±0.49 | 1.06 (-0.13) | 1.12±0.46 | 1.06±0.52 | 1.18±0.62 | 1.30±0.55 | 1.17 (-0.01) | 1.11 (-0.07) |
| Joint V2 | 1.03±0.62 | 0.85±0.51 | 1.87±0.84 | 0.96±0.58 | 1.18 (-0.01) | 1.19±0.56 | 1.32±0.73 | 1.27±0.85 | 1.34±0.67 | 1.28 (+0.10) | 1.23 (+0.05) |
| Joint V3 | 0.94±0.58 | 0.75±0.37 | 1.37±0.61 | 0.82±0.43 | 0.97 (-0.22) | 1.01±0.49 | 1.18±0.64 | 1.03±0.69 | 1.15±0.53 | 1.09 (-0.09) | 1.03 (-0.15) |
| Joint V1 (w/ CR) | **0.72±0.25** | 0.68±0.23 | 0.95±0.37 | **0.70±0.18** | 0.76 (-0.43) | 0.91±0.37 | 0.86±0.48 | 0.94±0.43 | 0.88±0.32 | 0.90 (-0.28) | 0.83 (-0.35) |
| Joint V2 (w/ CR) | 0.91±0.41 | 0.72±0.29 | 1.06±0.43 | 0.75±0.22 | 0.86 (-0.33) | 0.89±0.30 | 0.92±0.57 | 0.97±0.56 | 0.94±0.42 | 0.93 (-0.25) | 0.90 (-0.28) |
| **Ours (w/ CR)** | 0.87±0.29 | **0.58±0.17** | **0.84±0.32** | 0.72±0.24 | **0.75** (-0.44) | **0.83±0.36** | **0.56±0.23** | **0.85±0.39** | **0.83±0.37** | **0.77** (-0.41) | **0.76** (-0.42) |

Table 6: The performance of abdominal multi-organ segmentation by using 3D Transformer.

to optimization difficulties. Based on the above results, we use $\tau = 4$ in all subsequent experiments.

## Appendix F. Discussion

This paper aims to address cross-modal medical image segmentation based on the unpaired training samples, *e.g.,* CT and MRI images. Such multi-modal learning allows a single

| Methods | Abdominal CT | | | | | Abdominal MRI | | | | | Overall Mean |
|---|---|---|---|---|---|---|---|---|---|---|---|
| | Liver | Spleen | R-kdy | L-kdy | Mean | Liver | Spleen | R-kdy | L-kdy | Mean | |
| Dice Coefficient (avg.± std., %) ↑ | | | | | | | | | | | |
| Joint V3 | 94.6±2.2 | 93.3±1.8 | 88.9±2.3 | 88.7±3.5 | 91.4 | 93.8±2.3 | 88.5±2.7 | 92.7±1.5 | 91.5±3.1 | 91.6 | 91.5 |
| Joint V3 + $\mathcal{L}_{mcr}$ | 95.3±2.0 | 94.4±1.9 | 91.1±1.5 | 90.8±3.1 | 92.9 (+1.5) | 94.2±1.8 | 89.0±1.7 | 93.3±1.2 | 92.1±1.9 | 92.2 (+0.6) | 92.5 (+1.0) |
| Joint V3 + $\mathcal{L}_{icr}$ | 95.1±1.8 | 94.5±1.6 | 91.6±1.7 | 91.6±2.8 | 93.2 (+1.8) | 94.4±1.6 | 89.3±1.5 | 93.0±1.6 | 92.4±2.2 | 92.3 (+0.7) | 92.7 (+1.2) |
| Joint V3 + $\mathcal{L}_{mcr}$ + $\mathcal{L}_{icr}$ | 95.8±1.4 | 94.9±1.3 | 92.3±0.9 | 91.8±2.2 | 93.7 (+2.3) | 94.7±1.5 | 89.9±1.2 | 93.6±0.8 | 93.0±1.4 | 92.8 (+1.2) | 93.3 (+1.8) |
| Average Symmetric Surface Distance (avg.± std., mm) ↓ | | | | | | | | | | | |
| Joint V3 | 0.94±0.58 | 0.75±0.37 | 1.37±0.61 | 0.82±0.43 | 0.97 | 1.01±0.49 | 1.18±0.64 | 1.03±0.69 | 1.15±0.53 | 1.09 | 1.03 |
| Joint V3 + $\mathcal{L}_{mcr}$ | 0.91±0.46 | 0.67±0.31 | 0.98±0.52 | 0.77±0.37 | 0.83 (-0.14) | 0.96±0.43 | 0.79±0.45 | 0.98±0.66 | 1.04±0.49 | 0.94 (-0.15) | 0.89 (-0.14) |
| Joint V3 + $\mathcal{L}_{icr}$ | 0.88±0.37 | 0.65±0.28 | 0.93±0.43 | 0.74±0.33 | 0.80 (-0.17) | 0.91±0.35 | 0.74±0.38 | 0.89±0.51 | 0.96±0.45 | 0.88 (-0.21) | 0.84 (-0.19) |
| Joint V3 + $\mathcal{L}_{mcr}$ + $\mathcal{L}_{icr}$ | 0.87±0.29 | 0.58±0.17 | 0.84±0.32 | 0.72±0.24 | 0.75 (-0.22) | 0.83±0.36 | 0.56±0.23 | 0.85±0.39 | 0.83±0.37 | 0.77 (-0.32) | 0.76 (-0.27) |

Table 7: Ablation studies on abdominal multi-organ segmentation with 3D Transformer.

model to analyze data from multiple imaging devices, which greatly improves the efficiency of data usage. To better exploit the cross-modality information, we propose a novel method to accomplish cross-modal segmentation through learning structured semantic consistency between different modalities. Our model is designed for unpaired multi-modal images and is stable during the training phase. To learn structured semantic consistency across modalities (*i.e. the consistencies of semantic class representations and their correlations.*), we introduce a carefully designed External Attention Module (EAM) to conduct semantic consistency regularizations both at the modality and image levels. Such a module is very simple and flexible to use. It is only an external module used to embed cross-modal semantic consistency into the backbone network during the training phase, thus can be removed during the testing phase, ensuring the simplicity of the model. Moreover, the input of EAM is only the feature maps extracted at the specific scale. Therefore, it can be easily integrated onto various existing 2D and 3D Transformer architectures.

During the training process, we first construct globally learnable class embeddings for each modality, with the goal of capturing the representation of each class within each modality. Given that we use the same label taxonomy for unpaired CT and MRI, one intuitive strategy for learning consistent semantic information is to directly align class representations across modalities. However, we find that globally aligned class representations will not render the network more robust to sample variations. Driven by such a discovery, we further encourage the network to learn consistent semantic information at the image-level. The previous approach (Dou et al., 2020) directly align confusion matrices of predicted results across modalities. In contrast, we highlight semantic propagation at multiple scales from global to local, by interacting the global class representations across the entire dataset with the semantic features of each image, so as to learn image-specific class representations. Then, we derive multi-scale inter-class correlations within each image and dynamically establish its consistency between different modal data during training.

We conduct extensive evaluations on two medical image segmentation scenarios, outperforming the state-of-the-art methods with a large margin, *i.e.* 2.6% and 2.5% improvements on overall mean Dice for two tasks respectively. We further utilize few-shot setting to see how our method performs when one modality has far fewer samples than the other. Surprisingly, we find that a modality with a small number of training samples can boost the training of another modality with a large number of training samples, and a modality with a large number of training samples can greatly supplement the problem of another modality with a small number of training samples. And it is worth noting that since Transformer requires a large amount of data for training, we still initialize the model pretrained on Image-Net (Deng et al., 2009) for both datasets due to the limited data availability, otherwise, the performance will suffer.

A few limitations of the proposed method should be mentioned. Although the proposed method outperforms the state-of-the-art unpaired multi-modal learning schemes, the segmentation accuracy still has very huge room for improvement since we only use the basic Transformer architectures. This should be acceptable since we mainly focus on learning to align the structured semantic information from different modalities but not the detailed backbone network architecture design for pixel-level predictions. In addition, although the proposed method allows exploring more modalities (*e.g.,* CT, MRI, and X-ray) to learn semantic consistencies simultaneously. When the number of modalities increases, how to

align the semantic consistencies across these modalities is still not well addressed. One of the most straightforward ways is to group all modalities in pairs and align them one group by another. However, such a scheme may not the most efficient one when training the model. We plan to design a simple yet effective strategy to tackle the above issue in our future work.

Overall, we propose a novel scheme to learn structured semantic consistency between different modalities from unpaired samples via an attention mechanism. We apply it to the joint semantic segmentation of CT and MRI, whose appearances have a large discrepancy, and it achieves significant progress compared with counterparts. Intuitively, such a scheme can be easily extended to other domain alignment problems. For example, it can also learn unified abdominal organ representations from multiple datasets with different label taxonomy. This is common in practice, *e.g.,* some datasets label the `left kidney` and `right kidneys` as the same `kind`, while others label them as different semantic classes, or some datasets label the `intestines` as a single class, while others distinguish it with different segments. To tackle such an issue, we just need to introduce a learnable transformation matrix in our proposed EAM module to learn the mapping relationships between different semantic labels. We will explore these extensions in future work.

