# OpenReview forum: "Toward Unpaired Multi-modal Medical Image Segmentation via Learning Structured Semantic Consistency"
_MIDL.io/2023/Conference — MIDL 2023 Poster_

### Official Review · Reviewer_Tne6 · 2023-02-03

**Confidence:** 5
**Preliminary Rating:** 2

**Summary:**

The key contribution of this work is a transformer based approach for multi-modality segmentation. Two different public datasets are analyzed. The paper introduces an external attention module to the transformer architecture to enforce semantic consistency between the various segmented classes as well as other losses to enforce consistency of the extracted features between modalities. Although the paper cites some of the papers, several more recent works solving the problem of one to many modality transformation and segmentation are not presented, e.g. Jiang et.al's work using style encoding with VAEs applied to the CHAOS dataset in MICCAI 2020 as well as works by Zheng, Tan, Jiang, and Li who used a Transformer network for multi-modality segmentation (paper in PMB 2023 using multi-modality 3D Transformer network).

The method seems to provide incremental improvement over other baselines presented. No ablation experiments are presented so its unclear what is the contribution of the various losses added.

**Strengths:**

+ Transformer backbone improved with hierarchical attention for combining information from different modalities.
+ Modality specific and class specific correlations are derived that supposedly improve accuracy more than standard transformer architecture
+ Use of two public datasets to demonstrate feasibility

**Weaknesses:**

- The method seems overly complicated particularly given highly incremental improvements over other works. Only two prior works are presented, although several including some of the papers cited in this work have used these datasets to demonstrate accuracy improvements.
- There are no ablation tests so how the method really benefits from the various enhancements is unclear
- Semantic class reweighting and filtering looks like just the use of additional convolution kernels with attention module. How this specifically addresses the various semantic classes is not clear. Particularly, how do you enforce the consistency such as "left kidney should also have highest response in the region of the right kidneys" without actually coding it. If one has to code these type of rules, that requires inputing a lot of domain knowledge, which also seems problematic.
- Paper misses a lot of recent developments and restricts itself to papers from the same group.
- The method is very confusing to read. Please consider presenting the motivation and intuition behind the losses a little more clearly.

**Deanonymize Review:**

no

**Detailed Comments:**

Please see issues raised above.

**Paper Type:**

methodological development

**Questions To Address In The Rebuttal:**

Please clarify the improvements made in this work compared to more recent prior works such as by Jiang et.al in MICCAI 2020, and Zheng et.al

Please explain how semantic consistency is enforced for specific classes. The formulation looks to be generic attention module so how it actually enforces it for specific segmented structure is unclear.

The difference between modality-specific class embeddings and image-specific semantic correlations is not completely clear. They basically seem to be doing the same thing.

The accuracy improvements are highly incremental. There doesn't seem to be any significant differences. Was any significance testing done.

---

### Official Review · Reviewer_yXxz · 2023-02-04

**Confidence:** 2
**Preliminary Rating:** 5
**Recommendation:** Oral

**Summary:**

The authors present a multi-modal segmentation method that does not require image registration and leverages both image and semantic consistencies. The method is evaluated in two medical imaging segmentation scenarios: cardiac structure segmentation and abdominal multi-organ segmentation, achieving better dice scores and lower surface distances that single-modality training methods or joint trained methods.

**Strengths:**

This is a very complex and interesting method that uses a large combination of techniques. The language is correct. The experimental validation is correct and complete. Enough detail is presented and the language is correct.

**Weaknesses:**

None identified. The only comment one may rise is that it is hard to follow the work in an 8-page summary and a very extensive appendix. The work is probably best suited for a venue that allows for more space (i.e. journal) where the text can be re-arranged in a more straightforward manner.

**Deanonymize Review:**

no

**Paper Type:**

methodological development

**Questions To Address In The Rebuttal:**

It would be helpful to understand the number of training and test cases on the datasets.

The results would improve if statistical tests were made on the results. They seem to be far enough apart, and the variances of the proposed method are consistently lower than with previous methods, but understanding the statistical strength of the results would be helpful.

---

### Official Review · Reviewer_iXmo · 2023-02-06

**Confidence:** 3
**Preliminary Rating:** 4
**Recommendation:** Poster

**Summary:**

The author presents a semantic segmentation transformer model that learns the structural feature from unpaired multi-modal data via a plug-in External Attention Module. The major idea is, for CT and MRI images of the same class (e.g. liver), although the image appearance is different and the spatial features are not registered, the semantic structure are similar. Hence, the constraints for the model are from:

- Segmentation loss (Dice loss and CE loss).

- The similarity between the the (1) modality-specific class embeddings and (2) image-specific semantic correlation

The proposed method is evaluated on cardiac and abdominal datasets. The result outperforms the UMMKD in terms of Dice and surface distance.

**Strengths:**

- Learning correlations from multi-modality data is very useful to enlarge the deployment of deep models in medical image analysis due to the lack of annotation.
- The idea is straightforward and easy to implement.
- The experiment is conducted on multiple datasets and the result seems to be promising.


**Weaknesses:**

(1) Some of the descriptions/naming is confusing.

>- The author named the Q as 'modality-specific class embeddings' while the loss function L_{mcr} is leveraged to minimize the distance of Q_{M_1} and Q_{M_2}. Wouldn't Q be the class-specific embedding instead? Since the variance caused by the difference between modalities is the objective to be optimized.

(2) Input settings should be declaired.

>- Is it necessary to set the two input images in the same class? If not, some of the constraints seem to be invalid.

(3) More baseline Transformer models are needed.

>- e.g. Swin transformer (Liu et al)


**Deanonymize Review:**

no

**Paper Type:**

methodological development

**Questions To Address In The Rebuttal:**

(1) Should the input images be in the same class (e.g. liver)? If so, why the inter-class representation similarities are considered in Sec.2.2? If not, why the L_{mcr} should be minimized for two totally unrelated images?

(2)  Similarly, if the distance between the two modality embeddings Q_{M_1} and Q_{M_2} are to be minimized,  it seems to be contradictary to say 'each image shoudl have its own class representations due to appearance variance'.

---

### Meta-Review · Area_Chair_UtzN · 2023-02-23

**Recommendation:** Accept (Poster)
**Confidence:** 3

**Metareview:**

Learning correlations from multi-modality data was in general considered a strength of the paper by the reviewers. Another strength that was pointed out is the use of two public datasets and multiple baseline models for comparison. The reviewer who gave a lower rating noted that the method is very complex and ablation studies are missing. However, as the authors point out ablation studies are provided in supplemental material. There were other concerns brought up by the same reviewer which appear to be adequately addressed in the rebuttal (according to my judgment since the reviewer did not respond to the rebuttal). Criticisms about lack of clarity in certain places hopefully can be taken care of for the final version.